# Mechanosensitivity is an essential component of phototransduction in vertebrate rods

Ulisse Bocchero[1☯¤], Fabio Falleroni[1☯], Simone Mortal[1☯], Yunzhen Li[1], Dan Cojoc[2], Trevor Lamb[3], Vincent Torre[1,4,5]*

**1** Neurobiology Department, International School for Advanced Studies, Trieste, Italy, **2** Institute of Materials, National Research Council of Italy (CNR), Trieste, Italy, **3** Eccles Institute of Neuroscience, John Curtin School of Medical Research, The Australian National University, Canberra, Australia, **4** Cixi Institute of Biomedical Engineering, Ningbo Institute of Materials Technology and Engineering, Chinese Academy of Sciences, Zhejiang, China, **5** Center of Systems Medicine, Chinese Academy of Medical Sciences, Suzhou Institute of Systems Medicine, Suzhou Industrial Park, Suzhou, China

☯ These authors contributed equally to this work.
¤ Current address: Photoreceptor Physiology Group, NEI, NIH, Bethesda, Maryland, United States of America
* torre@sissa.it

**Data Availability Statement:** All relevant data are within the paper and its Supporting Information files.

**Funding:** The entire study was funded by the International School for Advanced Studies. VT

## Abstract

Photoreceptors are specialized cells devoted to the transduction of the incoming visual signals. Rods are able also to shed from their tip old disks and to synthesize at the base of the outer segment (OS) new disks. By combining electrophysiology, optical tweezers (OTs), and biochemistry, we investigate mechanosensitivity in the rods of *Xenopus laevis*, and we show that 1) mechanosensitive channels (MSCs), transient receptor potential canonical 1 (TRPC1), and Piezo1 are present in rod inner segments (ISs); 2) mechanical stimulation—of the order of 10 pN—applied briefly to either the OS or IS evokes calcium transients; 3) inhibition of MSCs decreases the duration of photoresponses to bright flashes; 4) bright flashes of light induce a rapid shortening of the OS; and 5) the genes encoding the TRPC family have an ancient association with the genes encoding families of protein involved in phototransduction. These results suggest that MSCs play an integral role in rods' phototransduction.

## Introduction

Photoreceptors are thought to be specialized cells devoted to the transduction of the incoming visual signals. Following strong illumination, rod outer segments (OSs) from mice [1] and fly photoreceptors [2,3] have been reported to increase their length and contract respectively, but rod OSs from frogs shrink their length by about 0.4–0.6 μm [4]. In addition, rod photoreceptors are known to shed old disks from their tip and to synthesize new disks at the base of the OS. These observations indicate the existence of mechanical machinery within rod OSs, but its action and role in phototransduction are completely unknown.

Mechanosensitive channels (MSCs) [5] have been found in olfactory sensory neurons [6] and possibly are expressed in many—if not all—neurons of the central nervous system. In

received the funding. The funders had no role in study design, data collection and analysis, decision to publish, or preparation of the manuscript.

**Competing interests:** The authors have declared that no competing interests exist.

**Abbreviations:** AM-ester, acetoxymethyl ester; BSA, Bovine Serum Albumin; CaSiR-1, Calcium ion detecting probe based on silicon rhodamine; CNGB, Cyclic Nucleotide-Gated ion channel Beta; CNR, National Research Council of Italy; FTL, Focused Tunable Lens; GC, guanylyl cyclase; GRK, G-protein receptor kinase; GsMTx-4, M-theraphotoxin-Gr1a; IR, infrared; IS, inner segment; Mb, megabase; MSC, mechanosensitive channel; Mya, million years ago; OOT, oscillatory optical trap; OS, outer segment; OTs, optical tweezers; QPD, quadrant photodetector; TRP, transient receptor potential; TRPC, transient receptor potential canonical; Tsat, saturation time; WB, western blot; 2R WGD, 2 rounds of whole-genome duplication.

bacteria, MSCs are thought to play a major role in maintaining osmotic equilibrium across their membrane especially upon hypoosmotic conditions; in these conditions, the opening of poorly selective MSCs contributes to the control of osmotic equilibrium [7–9]. MSCs in eukaryotic cells can be activated by light mechanical forces in the 10-pN range [10].

There are now several classes of ion channels implicated in the eukaryotic mechanotransduction machinery, including Piezo 1 and 2 channels [11] and transient receptor potential channels, referred as TRP channels. TRP channels form a superfamily of cation-selective ion channels located in cell membranes that are involved in various sensory modalities such as chemoreception, thermoreception, mechanoreception, and photoreception. TRP channels were discovered more than 3 decades ago in photoreceptors of fruit flies (*Drosophila*) [12–15]. The TRP superfamily in animals contains at least 7 families, comprising around 30 subfamilies, and the majority of these isoforms are expressed in vertebrates. Vertebrate TRPC channels (C for canonical) are so named because they are most closely related to the canonical TRP channels involved in *Drosophila* phototransduction. There are 7 subfamilies of vertebrate TRPCs, of which TRPC1, 4, and 5 are very closely related, as will be considered subsequently [14].

TRP channels are nonselective permeable cationic channels with a selectivity ratio $Ca^{2+}/Na^+$ that varies between the different family members [16]. Within the TRPC family, TRPC1 and TRPC6 have been reported to be activated directly by membrane stretch and curvature [17].

By combining electrical recordings, optical tweezers (OTs), and biochemical tools, we demonstrate in the present manuscript that 1) weak mechanical stimulation—of the order of 10 pN—applied briefly to either the OS or inner segment (IS) evokes a clear calcium transient; 2) inhibition of MSCs decreases the duration of photoresponses to bright flashes, and the magnitude of this effect increases with flash intensity; 3) bright flashes of light induce a rapid shortening—of the order of 200–300 nm—of the rod OS; and 4) the genes encoding the TRPC family of MSCs appear to have an ancient association with the genes encoding 3 families of protein that are directly involved in phototransduction in the rod OS. We also show that the MSCs TRPC1 and Piezo1 are present abundantly in rod ISs. Our analysis, together with 2 proteomic studies investigating the protein composition of the disks [18] and of the OSs [19], does not support their expression in OSs.

## Results

To investigate mechanosensitivity in rod photoreceptors, we decided to use OTs [20], which were used previously in our laboratory to trigger calcium transients in response to very weak forces in the 10-pN range [10]. Application of this approach requires the rods to be held in an environment of high mechanical stability, for example, lying on a rigid substrate. This makes it extremely difficult to simultaneously record their electrical responses, using either suction or patch pipettes. Instead, we chose to measure the functionality of rods through an infrared (IR) calcium dye, a Calcium ion detecting probe based on silicon rhodamine (CaSiR-1) [21]. Specifically, we loaded retinas using the acetoxymethyl ester (AM-ester) of this dye (see Methods), and then we mechanically dissociated individual rod photoreceptors and/or OSs. We viewed the preparation using IR illumination at 750 nm and an IR-sensitive video camera attached to the microscope. Then, from regions of interest, we recorded the fluorescence emitted by CaSiR-1 upon excitation with a 650-nm light (Fig 1A). This established that OSs lacking an IS fluoresced intensely (Fig 1A, left inset) and showed further that this fluorescence was not reduced by illumination with blue light. Thus, isolated OSs were unresponsive to blue light.

More nearly intact rods, in which the OS remained connected to at least part of its IS (i.e., OS + IS), fell into 2 categories. On the one hand, we found 1 category of "unresponsive" cells

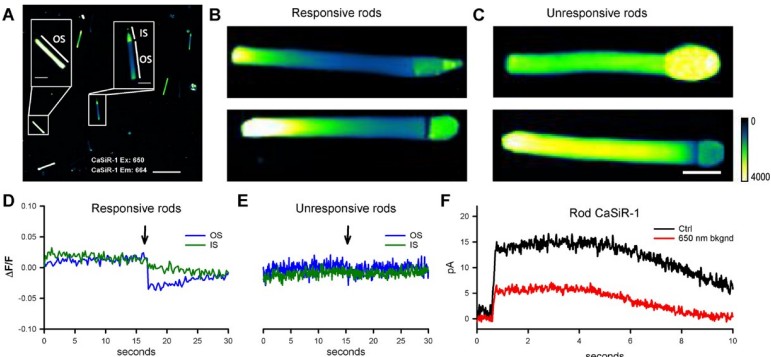

**Fig 1. Fluorescence emitted by the calcium-sensitive indicator, CaSiR-1, incorporated into rods isolated from the *Xenopus laevis* retina.** (A) Isolated rods in a dish, stained with CaSiR-1 excitation: 650 nm and emission: 664, showing a mixture of rods comprising both isolated OSs (left) and IS + OS (right) (scale bar, 50 μm). (B) Intact rods comprising an IS + OS typically display a pronounced longitudinal gradient of fluorescence, with the base of the OS quite dim. These 2 rods were responsive to blue light, as typified by panel D. The color scale from 0 to 4,000 is a linear scale. (C) Some ISs + OSs exhibited fairly uniform high fluorescence along the OS and were unresponsive to blue light (491 nm), as typified by panel E (scale bar is 10 μm for all the images). (D) Photoresponses from an IS + OS of the kind shown in panel B, from a ROI corresponding to 100 × 300 pixels of the OS (blue trace) and from an ROI corresponding to the IS (green trace). (E) No photoresponse was seen from the OS of ISs + OSs of the kind shown in panel C. (F) Suction pipette recordings in response to stimulation by bright blue flashes (approximately 2,500 photoisomerization [R*]) from an IS + OS loaded with CaSiR-1, in the absence (black trace) and in the presence (red trace) of the intensity of 650-nm light used for calcium imaging in the other panels. CaSiR-1, Calcium ion detecting probe based on silicon rhodamine; IS, inner segment; OS, outer segment; ROI, region of interest.

that showed no change in fluorescence upon exposure to blue light (Fig 1E). These cells were characterized by approximately uniform fluorescence along the OS, at a moderate to high level, and often showed a high level of fluorescence in the IS (Fig 1E, n>80). A second category of cells exhibited a marked gradient of fluorescence along the OS, with the basal section fluorescing only weakly (Fig 1A, right inset, and Fig 1B; *n* > 50). The emitted fluorescence of the CaSiR dye at the OS tip of IS + OS was usually 4 times larger than at its base so that the base appeared darker than the tip. Such cells exhibited a distinct drop in OS fluorescence in response to blue light (Fig 1D, blue trace; *n* = 10; see also S1 Fig). We hypothesize that this second category represents functional rods that exhibited a light-induced decrease in OS free intracellular calcium concentration as a result of the combination of activation of the phototransduction cascade and the existence of a circulating "dark current" driven by the ion gradients maintained by IS metabolism [22]. The longitudinal gradient in OS calcium concentration is likely to arise from the combination of 2 mechanisms: 1) the gradient of $Na^+$–$Ca^{2+}$–$K^+$ exchanger activity along the OS and 2) the gradient in $Na^+$ ion concentration caused by longitudinal diffusion of $Na^+$ ions towards the "sink" for intracellular $Na^+$ provided by Na-K-ATPase activity in the IS [23]. For the unresponsive cells (and for isolated OSs), it is plausible that there is a physical disconnect between the IS and OS that disrupts the maintenance of the required low $Na^+$ ion concentration in the OS.

For our experiments using OTs, we conclude from the results above that we can identify the functionality of rods lying on a rigid substrate: the cells need to have been preloaded with CaSiR-1, and then their emitted fluorescence is viewed upon excitation with 650-nm light. Functional rods display a pronounced longitudinal gradient of emitted fluorescence, with the basal end appearing dark.

In separate electrophysiological experiments we measured the effect of the 650-nm excitation light on the circulating current of functioning rods measured by presentation of a bright flash (Fig 1F). Compared with dark-adapted conditions (black trace), the circulating current in

the presence of the 650 nm excitation light (red trace) was reduced to about 40% (response amplitude 14 ± 2.5 pA in darkness and around 5.5 ± 2 pA with excitation; $n = 7$). From comparison with the effect of blue light, we conclude that the 650-nm light used to excite CaSiR-1 was approximately equivalent to about 500 R*/rod/s. Therefore, our measurements of changes in calcium concentration elicited by mechanical stimulation were performed during the equivalent of dim to moderate illumination of the rod, and we refer to this as "semi-dark-adapted" conditions.

## Mechanosensitivity of rods

Using the criteria developed above, we identified responsive isolated rods that had been loaded with the calcium-sensitive dye CaSiR-1, and we applied mechanical stimuli of approximately 10 pN to either the OS or the IS by means of an oscillatory optical trap (see S2 Fig and [10]). In semi-dark–adapted conditions, a mechanical pulse applied to a silica bead contacting the IS (Fig 2A) evoked a local increase in fluorescence (Fig 2B), with a magnitude Delta Fluorescence over the resting Fluorescence (DF/F) that could reach around 0.2 in 10–20 s (Fig 2C). The increase in fluorescence began with little delay from the mechanical stimulus (270 ms ± 80, $n = 12$), and the fluorescence signal remained localized to the IS, with no propagation to the OS (Fig 2B). Comparable results were obtained when the stimulating bead was touching the OS (Fig 2E): upon mechanical stimulation, the fluorescence increased locally (Fig 2F), reaching peak in about 10 s (Fig 2G), after a delay of no more than approximately 300 ms (Fig 2H). Collected results from 8 experiments on the IS and 8 experiments on the OS indicate a peak

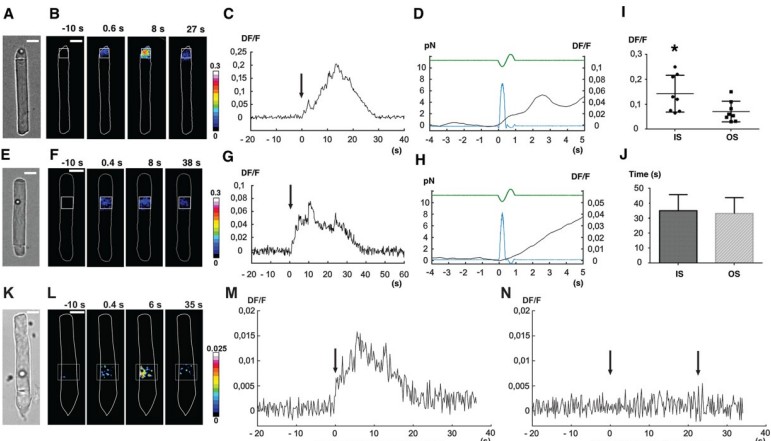

**Fig 2. Calcium response of *X. laevis* rods (OS + IS) to weak mechanical stimulation applied in the vertical direction (i.e., orthogonal to the plane of the images in A, E, K).** (A) A trapped bead in contact with the tip of the rod IS, under bright-field IR imaging. (B) Fluorescence change (DF/F) images, showing the ROI used to quantify the fluorescence change versus time. (C) Time course of the evoked DF/F change from the ROI in B. Mechanical stimulation was applied at time 0, as indicated by the arrow. (D) Trace from panel C on an expanded time base, additionally showing the FTL driving command (green trace) used to move the trapped bead, and the resulting force pulse (blue trace); the force applied to the IS was about 8 pN. (E, F, G, and H) As in A, B, C, and D but for mechanical stimulation of the rod OS. (I) Collected results for peak DF/F (mean ± SD) induced by mechanical stimulation for IS and OS, respectively (S1 Data). (J) Mean duration of calcium transients for IS and OS, respectively (S2 Data). Significance was determined with a two-tailed Student *t* test with $p < 0.05$. A calcium transient was detected when DF/F was above 0.004, approximately equivalent to 5-fold the background noise. Termination of detected transients was taken to occur when DF/F decreased below 0.004. (J, K, L, M, and N) As in E, F, and G but for mechanical stimulation before (M) and after (N) an exposure to GsMTx-4. In panels B, F, and L of Fig 2, the color maps indicate DF/F after spatial averaging over a window of 3 × 3 pixels. In subsequent panels (C, D, G, H, and M, N), DF/F was computed over the region indicated by the white boxes in panels B, F, and L. FTL, Focus Tunable Lens; GsMTx-4, M-theraphotoxin-Gr1a; IR, infrared; IS, inner segment; OS, outer segment; ROI, region of interest.

fluorescence increase (DF/F) of 0.14 ± 0.07 and 0.07 ± 0.04, respectively (Fig 2I and statistically different with $p < 0.05$), whereas the mean duration of these transients was around 35.1 s ± 10.7 in the IS and 33.2 s ± 10.6 in the OS (Fig 2L). The increase of intracellular calcium could have come across the plasma membrane or from internal stores or from both. In the case of the OS, "internal stores" encompasses possible fluxes from within the disks.

In several experiments, we repeated the same mechanical stimulation at least 3 times, and we observed a decline in the magnitude of the response (S3 Fig); it is possible that an initial fast component remained unchanged and that a second, larger component declined, but this is not clear from the results. To determine whether a similar phenomenon could have been initiated by attachment of the bead, we also conducted experiments monitoring the calcium signal along with bead position and force measurement during bead attachment (S4 Fig). We conclude from these experiments that when the bead attaches to the membrane, there is no major response or reaction in the cell.

In 3 experiments in which mechanical stimulation of the OS evoked a calcium transient, application of 10 μM M-theraphotoxin-Gr1a (GsMTx-4) abolished the response to subsequent mechanical stimulation (Fig 2N). These results indicate compartmentalization of calcium dynamics within the rod cytoplasm, and they suggest that rods are indeed mechanosensitive, i.e., that they express channels that can be activated by direct mechanical stimulation.

## Light-induced changes in rod OS length

Given that *Xenopus* rods respond to mechanical stimuli, we decided to test whether they also exhibited changes in OS length upon illumination, such as the shrinkage of about 0.4–0.6 μm reported by Lu and colleagues [4,24]. We chose to use OTs because of their high sensitivity, on the order of 1–10 nm, and rapid temporal resolution, in the millisecond range [20]; see also Methods and S4 Fig. We used the OT to position a 3.5-μm polystyrene bead above the tip of a rod OS and then gently lowered the bead until it made contact with the OS (Fig 3A), and then established good adhesion, as indicated by a decrease of the noise in the quadrant photodetector (QPD) trace [10]. Once the bead has sealed to the tip of the OS in this way, its precise

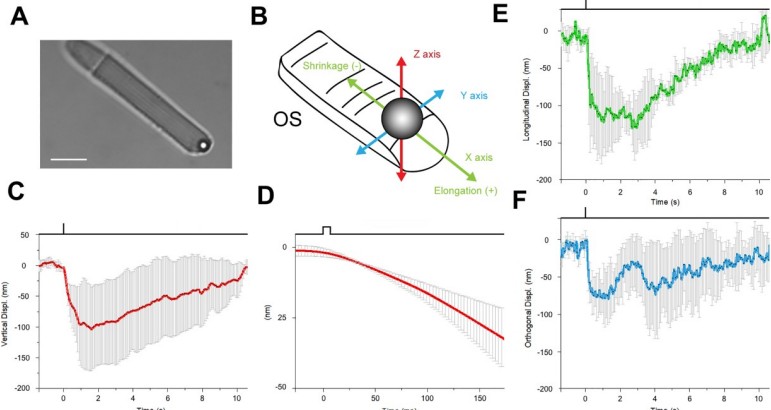

**Fig 3. Mechanical response of an *X. laevis* rod to light flashes.** The position of a bead sealed against the tip of the rod OS is monitored with OTs (see Methods). Following a bright flash of 491 nm, equivalent to about $10^4$ R*, a transient shrinkage is observed. (A) Bright-field IR image, showing a trapped bead in contact with the tip of the rod OS (scale bar, 10 μm). (B) Detail of the 3D tracking system. (C) Light-induced shifts in the *Z* axis of the trapped bead (downward is negative). (D) Expansion of the time base in E to examine the delay between light stimulus and bead movement. (E) Bead displacement along the direction of the rod OS (shrinkage is negative, and elongation is positive). (F) Bead displacement in the direction perpendicular to the rod OS axis. Data are representative of mean ± SD of 5 different experiments. IR, infrared; OS, outer segment; OTs, optical tweezers.

position (monitored by the OTs) provides a measure of the length of the OS so that any light-induced changes in OS length are recorded as displacements of the bead. We were able to measure the bead displacement in the X, Y, and Z directions, and we could then express the motion in terms of a longitudinal displacement in the X, Y plane along the direction of the OS (monitoring OS shrinkage or elongation) and a displacement along the vertical Z axis (see Fig 3B).

Fig 3 shows bead displacements in response to brief flashes delivering around 2,500 R*/rod. We consistently observed a light-induced shortening of the OS on the order of 100–200 nm in different experiments, as indicated by the significant shift in the axial and orthogonal position of the bead in contact with the tip of the OS after the onset of the light stimulus (Fig 3E and 3F). The delay to the onset of the bead displacement was about 50 ms (Fig 3D, mean 42.5 ms ± 12, $n = 5$). The shortening was transient, and in all experiments the bead returned to its original position within about 10 s, which is similar to the recovery of the electrical response to a flash of this intensity.

To avoid the possibility of artifacts caused by the OTs method, we decided to use conventional video imaging to measure the light-induced shrinkage of OS length. Thus, we turned off the IR laser used for optical trapping and used IR video imaging to visualize rods in pieces of retina (Fig 4). In this way, we could measure OS shrinkage under conditions in which the OS motion was not restrained by the adhesion with the substrate. We compared bright-field time-lapse images recorded before and after a stimulus, and we computed the kymographs (Fig 4B) along the segments indicated in A. Following bilinear interpolation (Fig 4C), we obtained the intensity profile of the rod tips before and following the bright flash of light (Fig 4D), from

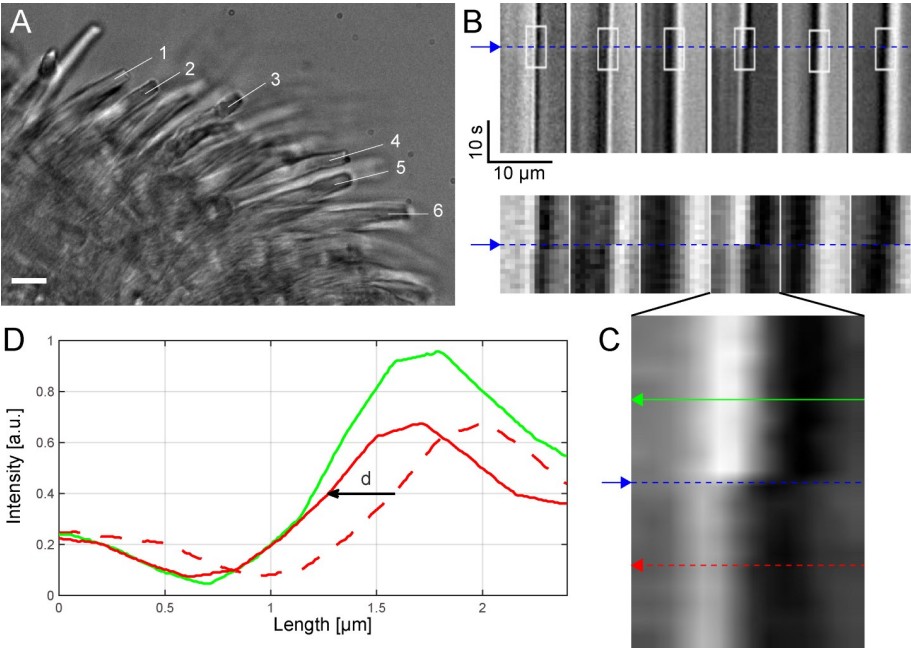

**Fig 4. Light-induced shrinkage of intact rods in pieces of retina.** (A) One frame from the movie (SI) showing a piece of retina with rods (scale bar, 10 μm). (B) Kymographs calculated along linear segments at the tip of the rods labeled with numbers from 1 to 6 in A show a shift, d: {d1 = 0.24, d2 = 0.32, d3 = 0.12, d4 = 0.31, d5 = 0.28, d6 = 0.33} μm in the intensity profile after the flash (blue line), confirming rod shrinkage. Small ROIs (white rectangles with size = 14 × 20 pixels) are selected and zoomed in the second row. (C) Bilinear interpolation of 1 ROI over a grid 20× bigger (280 × 400 pixels) than the original and 2 lines selected before and after the flash and used to calculate the shift d with subpixel precision. (D) Intensity profiles (green and dotted red) before and after flash. The shift d is calculated to minimize the difference between the intensity profiles (solid red and green lines) (S3 Data). a.u., arbitrary unit; ROI, region of interest.

which we calculated the OS shrinkage with subpixel resolution (see Methods) and found it varying from 120 up to 330 nm. OTs provide higher spatial resolution measurement of the time course of the light-evoked OS shrinkage but under conditions in which OS are restrained at some extent. Live cell imaging allows a measurement of OS shrinkage under more physiological conditions but at a much lower temporal resolution, i.e., of 500 ms.

The combination of these 2 approaches confirms that flashes delivering approximately 2,500 R*/rod trigger a rapid transient shortening—within 10–20 ms from the flash delivery— of *X. laevis* rods, similar to that reported recently in [4,24], in which it was referred to as transient retinal phototropism. The comparison of the rising phase of the photoresponse and of the time course of OS shortening shows that these 2 processes are fast, occurring with a similar delay from the onset of the light flash (see S5 Fig).

## Rod photoresponses in the presence of MSC inhibitor

Given that rod photoreceptors show mechanosensitivity, we investigated the role of MSCs in phototransduction by recording rod photocurrents with a suction pipette and then applying the MSC inhibitor GsMTx-4 [25]. GsMTx-4 is a small peptide obtained from a spider venom and has been shown to inhibit several MSCs from both the Piezo and TRP families [26,27]. It is thought to act at the interface between the lipids in which the MSC is embedded [28], thereby reducing the effective magnitude of the mechanical stimulus acting on the MSC gate; thus, GsMTx-4 is a gate modifier rather than a specific ion pore blocker. According to Gnanasambandam and colleagues [28], GsMTx-4 is stabilized by lysine residues and occupies a small fraction of the surface area in unstressed membranes. When applied tension reduces lateral pressure in the lipid phase, those residues penetrate deeper, acting as "area reservoirs," leading to partial relaxation of the outer monolayer, thereby reducing the effective magnitude of the stimulus acting on the gate of the MSC.

We delivered GsMTx-4 using a second similar pipette connected to a picospritzer and positioned 50–100 μm from the OS of the recorded rod (Fig 5A). Prior to drug exposure, presentation of bright flashes of about 2,500 R*/rod [29] triggered suppression of the rod circulating current for 5 s (Fig 5B and 5C). When the inhibitor was gently injected into the bath, using around 4 psi of pressure, the OS of the recorded rod was displaced from its original position (compare upper and lower panels in Fig 5A), signaling the arrival of GsMTx-4. The same illumination then elicited photoresponses of shorter duration (compare black and red traces in Fig 5C; *n* = 13). Subsequently, after the injection of GsMTx-4 had been terminated, the photoresponses recovered their original time course (compare black and blue traces). When the same experiment was repeated in the absence of GsMTx-4 in the pipette but with a similar degree of OS displacement, no significant shortening of photoresponses was observed (Fig 5D and 5E; *n* = 15).

We analyzed the effect of GsMTx-4 on photoresponses to flashes with intensity ranging from dim (5 R*/rod) to bright (2,500 R*/rod); compare black and red traces in Fig 5F and 5G (*n* = 10 rods). Application of the MSC inhibitor had negligible effect on photoresponses to dim flashes (*n* = 7–10; Fig 5H), and it had relatively little effect for flashes of intermediate intensity (Fig 5F, 5G and 5I). It was only for flashes of saturating intensities (i.e., greater than approximately 100 R*/rod) that the time course was shortened by GsMTx-4, and for these saturating flashes, the magnitude of the response shortening increased with increasing flash intensity (see Fig 5I). We also analyzed photoresponses to steps of light lasting 20 s in control conditions and in the presence of GsMTx-4, and the rising and falling phases of these photoresponses were rather similar (see S6 Fig). Although the GsMTx-4 trace is very slightly smaller than the control, this might readily be explicable in terms of a slight rundown in the cell's circulating current.

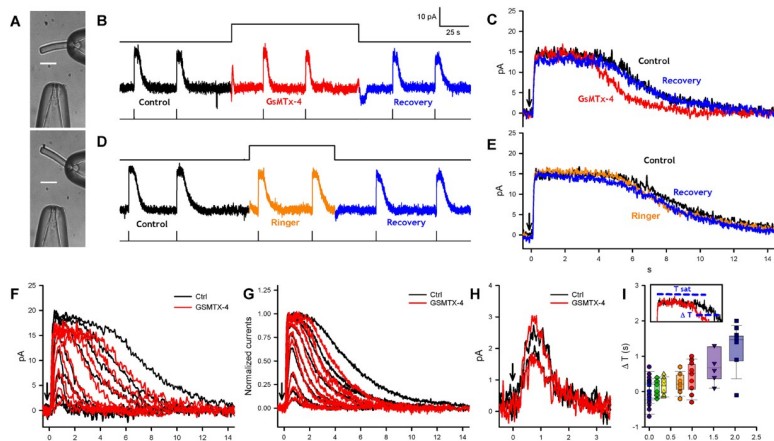

**Fig 5. The effect of MSC blocker GsMTx-4 on photoresponses from *Xenopus* rods.** (A) IS of a rod trapped in a recording pipette and a glass pipette connected to a picospritzer, containing GsMTx-4. Upon the activation of the picospritzer, the rod OS was tilted and returned to its initial position (scale bar, 20 μm). (B) The photocurrent elicited by flashes of about 2,500 R*/rod before, during, and after an exposure to GsMTx-4 lasting 120 s. (C) Exposure to GsMTx-4 (red trace) shortens the duration of the bright-flash photoresponse by more than 1 s compared with those obtained immediately before (black) and immediately after (blue) exposure to GsMTx-4; mean duration (measured as the difference in time between 50% of the falling phase and 50% of the rising phase of the response) 4.7 ± 1.8 s in control and 3.4 ± 1.3 s in GsMTx-4 ($p < 0.001$). (D and E) As in B and C, but the picospritzer injected Ringer solution. Mean duration 6.2 ± 1.7 s in control and 5.9 ± 1.9 s in Ringer (ns). (F) Comparison of photoresponses in control conditions (black traces) and in the presence of GsMTx-4 (red traces) for 1 cell exposed to flash of approximately 5, 10, 25, 50, 100, 250, 500, 1,000, and 2,500 R*/rod. (G) As in F, but in this case each trace was averaged over photoresponses obtained from 6–7 different rods. The amplitude of the maximal photoresponse was normalized to unity for recordings both in Ringer (black traces) and in the presence of GsMTx-4 (red traces). (H) The effect of GsMTx-4 on dim flash photoresponses for one cell; the flash intensities were 5 and 10 R*/rod ($n = 6$–7). (I) Relation between the GsMTx-4-induced shortening (ΔT) of photoresponse duration and the Tsat of the response. For 250, 500, 1,000, and 2,500 R*/rod, the shortened time courses were of 0.3 s, $p < 0.05$; 0.5 s, $p < 0.01$; 0.7 s, $p < 0.05$; and 1.3 s, $p < 0.001$, respectively. In all experiments, the concentration of GsMTx-4 in the picospritzer pipette was 5 μM, we estimate that at the rod OS was in the micromolar range; $n = 6$–7 (S4 Data). GsMTx-4, M-theraphotoxin-Gr1a; IS, inner segment; MSC, mechanosensitive channel; OS, outer segment; Tsat, saturation time.

## Relationship of the *TRPC1* gene to the genes underlying vertebrate phototransduction

We examined gene synteny for both *TRPC1* and *PIEZO1*, and we discovered that the *TRPC1* gene is closely associated with several genes that encode proteins involved in the vertebrate phototransduction cascade. In particular, *TRPC1* is clearly a member of the paralogon that comprises the visual GRKs, the arrestins, and the visual GCs (guanylyl cyclases). We also found suggestive evidence that *PIEZO1* and *PIEZO2* may be located on another paralogon that includes the Cyclic Nucleotide-Gated ion channel Beta 1 (*CNGB1*) and *CNGB3* genes that encode cyclic nucleotide-gated ion channel (CNGC) β-subunits; this possibility deserves future examination.

The syntenic relationship between the TRPCs and the 3 other families of genes mentioned above is summarized in Fig 6; note that, for purposes of illustration, we have chosen to show just 4 families and just 2 species from the larger set presented in S8 Fig. Each column represents the remaining members of the quartet of genes that were generated from a single ancestral gene through 2 rounds of whole-genome duplication (2R WGD) in a protovertebrate organism some 500 million years ago (Mya). Each row shows a region of either 1 or 2 chromosomes in an extant organism, and where 2 regions are shown, this is presumed to be the result of chromosomal rearrangements over 500 My. Examination of the larger data set in S7 Fig

**Fig 6. Summary of synteny for 4 gene families from 2 species.** Gene locations are shown for TRPCs, visual GRKs, visual GCs, and arrestins. Number under each gene name represents the gene position in Mb on the indicated chromosome. Note that there is strong evidence that each interrupted row (where there is a break in continuity of a chromosome) corresponds to a contiguous set of genes in the ancestral quadruplicate genome. See S7 Fig for synteny across 11 gene families and 4 species. GC, guanylyl cyclase; GRK, G-protein receptor kinase; Mb, megabase; TRPC, transient receptor potential canonical.

provides powerful evidence that each of the 4 rows is contiguous and represents the current rearrangement of genes on the 4 ancestral chromosomes that existed shortly after 2R WGD.

In Fig 6, the proximity of TRPC family members to members of the other 3 families is impressive. For example, the distance from *TRPC1* to *GRK7* is just 0.2 megabases (Mb) in both spotted gar (Fig 6A) and chicken (S7B Fig) and is <1 Mb in both human (Fig 6B) and opossum (S7C Fig). Likewise, the distance from *TRPC5* to GC-F (= *GUCY2F*) is <1 Mb in spotted gar (Fig 6A) and <2.5 Mb in human (Fig 6B); in the other 2 species, the loss of the gene for GC-F precludes this comparison (S8 Fig). Furthermore, as shown previously [30], in several cases members of the other 3 families of phototransduction genes are close to each other; for example, *GRK1B*, GC-E, and *ARRB2* are close to each other in spotted gar (Fig 6A). Such proximity in extant chromosomes is an important telltale sign of ancient proximity because of the very low likelihood that random chromosomal rearrangements could bring so many genes into mutual proximity; instead, random rearrangements are likely to obscure any proximity that originally occurred. Therefore, we conclude that it is very likely that in a protovertebrate organism, the ancestral genes (TRPC, visual GRK, visual GC, and arrestin) were arranged in close proximity to each other prior to quadruplication during 2R WGD.

## Localization of MSCs in *X. laevis* rods

To better investigate the expression of MSCs Piezo1 and TRPC1 in rods, we stained retinas by immunofluorescence with antibodies for Piezo1 and TRPC1 (see Methods). Immunolabeling for Piezo1 (Fig 7A) shows a clear staining in rods, with a punctate expression in the ellipsoid region of the IS. On the other hand, in Fig 7B, the apparent labeling for TRPC1 along the OS may in fact represent autofluorescence associated with the high density of rhodopsin

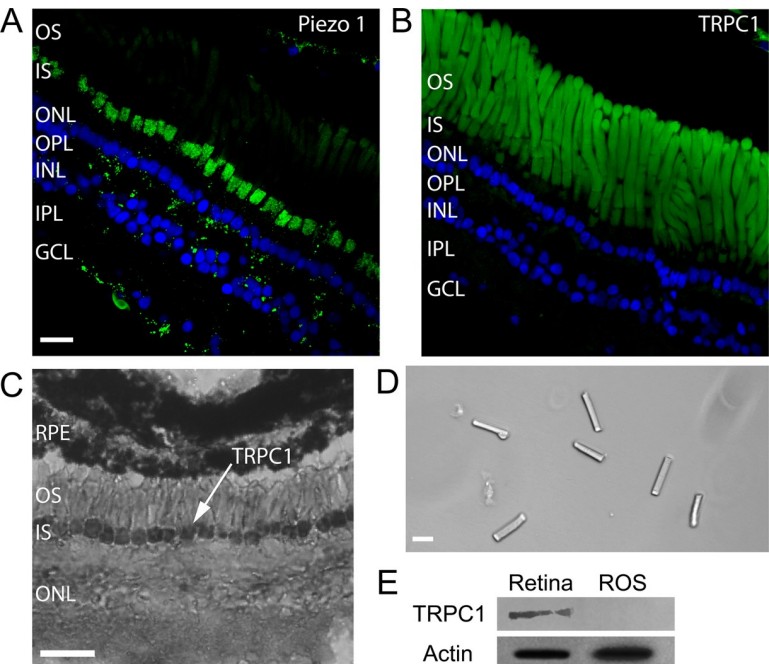

**Fig 7. Expression of mechanosensitive channels in the *X. laevis* retina.** (A) Immunofluorescence for Piezo1 in green and DAPI in blue. (Scale bar is 20 μm for images A, B, and D.) (B) Immunofluorescence for TRPC1 in green and DAPI in blue. (C) Immunohistochemistry for TRPC1. (Scale bar, 50 μm.) (D) Isolated OS obtained by sucrose centrifugation. (E) WB for TRPC1 from the whole retina and isolated ROS as those shown in D. GCL, Ganglion Cell Layer; INL, Inner Nuclear Layer; IPL, Inner Plexiform Layer; IS, inner segment; ONL, Outer Nuclear Layer; OPL, Outer Plexiform Layer; OS, outer segment; ROS, Rod outer segment; RPE, Retinal Pigment Epithelium; TRPC, transient receptor potential canonical; WB, western blot.

molecules [31]; accordingly, these data cannot be taken to show definitive expression of TRPC1 in rod OSs.

Molnar and colleagues [32] have shown that TRPC1 is expressed mostly in rod ISs in mouse, using RNA in situ hybridization with nitroblue tetrazolium. To get clearer evidence on the expression patterns of TRPC1, we decide to perform immunohistochemical staining of *Xenopus* retinas with the same antibody used for the immunofluorescence. Indeed, immunohistochemical staining confirmed that TRPC1 is expressed primarily on the IS membrane of rod photoreceptors (Fig 7C) with possible weaker staining also in the OSs. Moreover, we isolated rod OSs (Fig 7D), IS + OS, by purification on an OptiPrep gradient, [33] and we performed standard western blot (WB) with antibodies for TRPC1 (Fig 7E). This analysis shows that TRPC1 and Piezo1 channels are abundantly present in the retina and in ISs, but not in the OSs (Fig 7D). This conclusion is in agreement with the conclusions of previous proteomic studies [18,19], which did not report the presence of Piezo and TRP channels either in OSs or in disks. We were not able to determine the presence or absence of TRPC1 and Piezo1 channels in a population of isolated and purified disks by WB analysis.

## Discussion

The present manuscript shows a number of novel, to our knowledge, features of rod photoreceptors involving mechanosensitivity and mechanosensitive channels. TRPC1 and Piezo1 MSCs are present in rod photoreceptors; in addition, it has long been known that the actomyosin complex is associated with the ciliary machinery linking the light-sensitive OS to the IS

[34]. Interestingly, weak mechanical stimulation—on the order of 10 pN—applied briefly to either the OS or IS evokes clear calcium transients, which are inhibited by the toxin GsMTx-4. Moreover, the inhibition of MSCs through GsMTx-4 decreases the duration of photoresponses to bright flashes, and the magnitude of this effect increases with flash intensity. The genes for the TRPC family appear to have an ancient association with 3 other families of genes that are directly involved in phototransduction in the rod OS. These results suggest that MSCs play an integral role in the regulation of rod phototransduction.

Both TRPC1 and Piezo1 channels are multimodal and have been reported to be gated and modulated by temperature and second messengers. The observation that very weak mechanical stimulation of the IS elicits a transient increase in intracellular calcium concentration is consistent with the view that these ionic channels in the IS are mechanosensitive. Mechanosensitivity in OSs seems to have a more complex origin; weak mechanical stimulations evoke brief calcium transients (Fig 1), but we have not been able to determine in a conclusive way the presence of TRPC1 and Piezo1 and 2 channels in the OS. It is possible, however, that in the OS there are additional MSCs or that mechanosensitivity has a different origin: it is conceivable, indeed, that the small indentation occurring during the applied mechanical stimulations—on the order of some hundreds of nanometers—disrupts disks known to be filled by calcium ions [35] and therefore induces a localized transient calcium increase. In some experiments (see S5 Fig), we observed that in the presence of GsMTx-4, mechanical stimulation did not evoke any calcium transients, but spontaneous calcium transients could be observed. This observation suggests that calcium transients could occur in the absence of any apparent mechanical stimulation and that mechanical stimulations could modulate the frequency of these transients. In conclusion, we are confident that we have identified the molecular origin of mechanosensitivity in the IS, but mechanosensitivity in the OS could involve both MSCs and a direct mechanical action on the disks.

We observed that GsMTx-4 caused a shortening of the duration of bright-flash responses (Fig 5B and 5C). The exposure to GsMTx-4 does not induce any measurable change in the amplitude of the saturating current (Fig 5C), indicating either that MSCs are not activated in dark-adapted conditions (i.e., before the exposure to the bright flash) and/or that the ionic current flowing through MSCs is small and cannot be easily measured. On the basis of these observations, we suggest that in the presence of GsMTx-4, a bright flash results in a more pronounced light-induced drop in calcium concentration because the MSCs are inhibited. Thus, the inhibition of MSCs will block an influx of calcium into the cytoplasm that is normally stimulated by mechanical movement of the OS triggered by the bright flash and thereby result in a larger decline in free calcium concentration. The ensuing shortening of the bright-flash photoresponses could result from an effect of the lowered calcium concentration via either increased cyclase activity [36] or decreased $R^*$ lifetime [37,38] or both.

If MSC channels are activated during phototransduction, a key issue is what mechanical stimulation activates them? Three possibilities spring to mind: activation of the actomyosin complex; a drop of intracellular osmotic pressure caused by the transient abolition of the photocurrent, and dimensional changes such as the light-induced shortening of the rod OS (Fig 3 and [4,24]). We suggest that the first of these is unlikely; although the IS is rich in actin, in the OS actin is present only at its base and not in the whole OS.

We have not been able to estimate the magnitude of any light-induced change in intracellular osmotic pressure, though we expect it will be small. Although suppression of the dark current of 50 pA corresponds to a reduction in the entry into the OS of around $1.5 \times 10^8$ monovalent cations per second, this is counterbalanced by an equal reduction of current flow out of the OS and into the IS [39]. Therefore, in response to a saturating flash of light, it is possible that a drop in osmolarity develops, but it seems likely that the reduced efflux of positive charge from the OS to the IS through the ciliary neck may minimize this effect.

In agreement with previous observations [4,24], we confirm that bright flashes of light elicit a transient shortening of the rod OS. These flash-induced movements begin with a delay around 50 ms for a flash of about 2,500 R*/rod (Fig 3). This transient shortening of OSs, also referred as transient retinal phototropism, is thought to be associated with early, disk-based stages of the phototransduction cascade [4] and is not caused by the light-induced suppression of the photocurrent. Transmission electron microscopy shows that the shrinkage is associated with a decrease in the space between disks rather than any change in thickness of the disks themselves [4], and this decrease in OS cytoplasmic volume will necessarily cause an increase in osmotic pressure. Hence, we propose that activation of MSCs is elicited either directly by the change in interdisk spacing (especially if MSCs are located in the disk membranes) or secondarily by the change in cytoplasmic osmotic pressure. We are aware, however, that this reduction of interdisk spacing will initiate adjustments of the hydrostatic pressure and of water volume that have to be properly addressed and understood.

The transient shortening of rod OS is likely to play a major role in phototransduction, which is, at the moment, not entirely understood. This shortening is associated with the early stages of phototransduction occurring within some tens of milliseconds following rhodopsin activation [4], and it is not clear how the biochemical cascade initiated by photon absorption leads to a shrinkage of the interdisk space; we believe that this shrinkage represents a missing step for a complete and full understanding of phototransduction.

Although it is widely thought that sensory neurons (such as photoreceptors) are specialized to transduce just a single sensory modality, the present investigation not only shows that rods express MSCs but also demonstrates that they display mechanosensitivity. We hypothesize that rod photoreceptors require such mechanosensitivity both for the optimal operation of the phototransduction machinery and for the maintenance of cellular integrity.

## Methods

### Ethics statement

All the experiments described in this manuscript were performed in accordance with the guidelines of the International School for Advanced Studies ethics committee and according to the Italian and European procedures for animal care (d.l. 116/92; 86/609/C.E.).

### Immunofluorescence

Retinas were fixed with 4% paraformaldehyde for 60 min at room temperature, followed by permeabilization with PBS plus 0.1% Triton X-100, blocked with 1% BSA (Bovine Serum Albumin) and incubated overnight with primary antibodies anti-Piezo1 (1:300) or anti-TRPC1 (1:300) from Alomone Labs (Jerusalem, Israel). Retinas were then washed with cold PBS 3 times for 5 min each and incubated with Alexa 488-labeled goat anti-mouse secondary antibody (1:400) or Alexa 594-labeled goat anti-rabbit secondary antibody (1:400) and actin (phalloidin) (1:50) at room temperature for 1 h and then stained with Hoechst (all from Life Technologies, Carlsbad, CA, USA). Retinas were examined with a NIKON A1R confocal microscope (Nikon, Tokyo, Japan) equipped with 405, 488, and 561 excitation lasers, 40× objective (NA 0.75) and 60× oil immersion objective (NA 1.40).

### Isolation of photoreceptors and electrical recordings

The eyes of *X. laevis* frogs were enucleated and hemisected under a stereotactic microscope with an IR 820 nm illumination. Dissociated rods were obtained as reported previously [29]. Briefly, the intact rods obtained by mechanical dissociation were immersed in Ringer solution

containing 110 mM NaCl, 2.5 mM KCl, 1 mM CaCl2, 1.6 mM MgCl2, 3 mM HEPES-NaOH, 0.01 mM EDTA, and 10 mM glucose (pH 7.7–7.8, buffered with NaOH). All chemicals were purchased from Sigma-Aldrich (St. Louis, MO, USA). All experiments were performed between 22˚C and 24˚C and images acquired using HCImage software 4.3.1.33 (Hamamatsu Corporation, Bridgewater, NJ, USA).

S1 Fig shows representative images of rods following mechanical dissociation in bright field (B) and in fluorescence (A and C) emitted by our dye CaSiR-1. More than 30% of the isolated rods are composed of an IS and OS where the IS is bright and the base of the OS is dark and its tip more luminous. These ISs + OSs are functional rods because they exhibit a light response (measured as a change in fluorescence).The quality of the preparation varies with the animal and the skill of the experimenter, as shown in panel E of the above figure, but most IS + OS rods are functional, varying from 75% up to 100%.

From our previous experience, toad rods (*Bufo marinus* and *B. bufo*) are rather fragile, but rods from the tiger salamander (*Ambystoma tigrinum*) are much more robust. We have observed that rods from *X. laevis* frogs, properly dissociated, are often functional. In addition, *X. laevis* frogs are easier to keep than tiger salamanders, and they also reproduce well in captivity. These are the reasons we have opted to work with *X. laevis* rods.

After mechanical isolation, electrical recordings were obtained as described in [24]. Rods were viewed under 900-nm light using 2 cameras (Hamamatsu ORCA-Flash 4.0; Hamamatsu Corporation and Jenoptic ProgRes MF; JENOPTIK I Optical Systems, Goeschwitzer, Jena, Germany) at 2 magnifications and stimulated with 491-nm diffuse light (Rapp OptoElectronic, Hamburg, Germany) from the ×10 objective of an inverted microscope (Olympus IX71; Olympus Corporation, Tokyo, Japan). Photoresponses were recorded using an Axopatch 200A (Molecular Devices, San Jose, CA, USA) in voltage-clamp mode. The current was digitized at 10 kHz and low-pass filtered at 20 Hz. All recordings were processed, analyzed, and baseline corrected with Clampfit 10.3 (Molecular Devices).

## Calcium imaging

Retinas were loaded with a cell-permeable calcium dye CaSiR-AM (Life Technologies) and Pluronic F-127 20% solution in DMSO (Life Technologies) at a ratio of 1:1 in Krebs-Ringer's solution containing 119 mM NaCl, 2.5 mM KCl, 1 mM $NaH_2PO_4$, 2.5 mM $CaCl_2$, 1.3 mM $MgCl_2$, 11 mM D-glucose, and 20 mM HEPES (pH 7.4) at 37˚C for 45 min. After incubation, the isolated rods were washed 3 times for at least 15 min total to allow complete intracellular de-esterification of the dye, then transferred to the stage of an Olympus IX-81 inverted microscope equipped with LED illumination (X-Cite XLED1 from Excelitas Technologies, Waltham, MA, USA). The experiments were performed at room temperature (between 22˚C and 24˚C), and images were acquired using Micromanager software with an Apo-Fluor 60×/1.4 NA objective at a sampling rate of 5 Hz for 3–10 min.

Changes of intracellular calcium were quantified by computing DF/F = (F($t = 0$)–F($t$))/F(0), where F(0) is the fluorescence intensity at the beginning of the experiment and F($t$) is the fluorescence intensity at time $t$. The fluorescence analyzed in panels C, D, G, H, M, and N of Fig 2 was computed in the large region indicated by the white boxes in Fig 2B, 2F and 2L. In panels B, F, and L, the color maps indicate DF/F computed at each pixel within the white box after a spatial averaging in a window of $3 \times 3$ pixels.

## Mechanical stimulation using the oscillatory optical trap

To mechanically stimulate the cell, we used a polystyrene bead with a diameter d = 3.5 μm (G. Kisker GbR, Steinfurt, Germany) optically manipulated in an oscillatory optical trap (OOT)

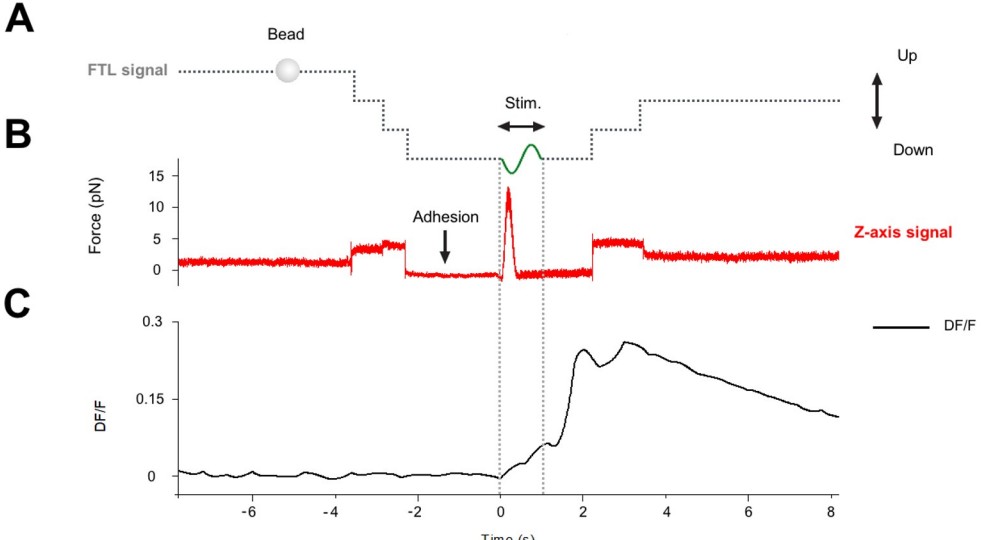

**Fig 8. Analysis of the recording of the QPD trace and of DF/F.** (A) Voltage command to the FTL, controlling the vertical displacement of the optical trap. The steps in the dotted lines show the lowering of the bead before adhesion and raising of the bead after mechanical stimulation. (B) Force recorded in the vertical direction by the QPD. (C) Fluorescence signal DF/F monitoring the level of intracellular calcium. There was no change in DF/F at the time of the adhesion of the bead (marked by a decrease of noise in the force trace), but DF/F increased following the mechanical stimulation, indicated by "Stim." FTL, Focused Tunable Lens; QPD, quadrant photodetector.

(S2 Fig). The main component of the OOT is the Focused Tunable Lens (FTL; EL-10-30-NIR-LD, Optotune AG, Dietikon, Switzerland), of which the focal length can be precisely tuned to change the vertical position of the trapped bead. Cell stimulation is achieved by trapping the bead above the cell and then moving it against the cell membrane. A complete description of all components of the setup are found in [10]. The setup can be used for measuring displacements (Fig 3) and for applying forces (Fig 4), and the shift between the 2 modes of operation is obtained by inserting the FTL into the optical path of the laser when applying forces (see S2 Fig).

Analysis of the recording of the QPD trace and of DF/F (see Fig 8) allows us to determine the moment in which the bead attaches/seals to the membrane and this event is not associated with any change of DF/F. The moment in which the bead seals to the membrane is revealed by a decrease of the noise in the QPD recording, in a way reminiscent of what occurs when a patch pipette seals on the membrane in electrophysiological experiments. We conclude from these experiments that when the bead attaches to the membrane, there is no major response or reaction of the cell.

### Measurement of the rod shrinkage by bright-field time-lapse imaging

We used an objective lens with 60× magnification, numerical aperture NA = 1.4, and a Hamamatsu ORCA-D2 camera with a sensor pixel size of 6.5 μm. Considering the light wavelength, λ = 750 nm, the lateral optical resolution is r of approximately 320 nm. The sensor resolution is given by the pixel size. We used binning 2×; thus, 1 image pixel corresponds to about 220 nm on the sample ($p = 6.5 \times 2 \times 1{,}000/60 = 216.7$ nm to approximately 220 nm). The sensor resolution matches the optical resolution at the limit $3p$ is approximately $2p$ (matching condition: $3p \geq 2p$). The experiment consists in recording the image of a piece of retina before and after a flash of light at 2 frames/s and observing potential retraction of the rods. One frame illustrating a piece of retina with rods is shown in Fig 4A. We measure the retraction of the

rods indicated in figure with numbers from 1 to 6. For each rod, we define a segment of the same length (L = 10 μm) along the rod axis (white lines), and we calculate the kymograph. The kymogaph is a graph space versus time, showing how the intensity distribution along the segment varies in time (Fig 4B). This variation is directly related to a potential retraction of the rod. As one can observe from the kymograph, after the flash (indicated by the blue line), there is a clear shift of the gray value, Δ, corresponding to the retraction of the rod. We calculate the values for the shift Δ with subpixel precision. To do this, we first select small regions of interest (white rectangles in Fig 4B) of size ROI = 14 × 20 pixels. Then, using bilinear interpolation, we define a new ROI over a grid with 20 times more points (pixels) in each direction than that of the original ROI (Fig 4C). We then plot the intensities along the green and red lines, corresponding to the darker and lighter regions (after flash). To determine the shift Δ, we calculate the quantity: $A = \int [(Id(x - \Delta) - Il(x))]^2 dx$, where $Id(x)$ and $Il(x)$ represent the intensity profiles for dark and light regions and $\Delta = k \times p/20$, with k = 1:40. The subpixel value of the shift Δ is given by k for which the quantity A is minimized (Fig 4D). We found that the rods shifted by Δ > 240 nm (i.e., more than 1 pixel), except rod 3, for which Δ = 120 μm. The maximum shift Δ max = 330 nm corresponds to rod 6.

## Data and statistical analysis

For calcium experiments, the DF/F values were obtained through custom MATLAB (The MathWorks, Natick, MA, USA) code and the ImageJ software v1.6 (National Institutes of Health, Bethesda, MD, USA). All results are presented as mean ± SD, and significant differences were determined using a *t* test with $p < 0.05$ (GraphPad Prism 7, GraphPad software, San Diego, CA, USA). For electrophysiological experiments, the parameters of rod responses were analyzed with Clampfit 10.3, and the statistical significance was determined using the paired *t* test in SigmaPlot 13.0.

## Supporting information

**S1 Fig. The quality of the preparation.** (A and C) Images of the fluorescence emitted by the calcium fluorescent dye CaSiR from rods in the preparation. (B) is a bright-field image obtained with a light source at 750 nm corresponding to the fluorescent image in A. Rods were viewed with a 60× objective, and of the 4 rods viewed in B, 3 were composed of IS + OS (labeled 1, 2, and 3). (D) Photoresponses obtained from the IS + OS shown in A and C. The blue traces are the light stimulus monitor; flash intensity was equivalent to about $10^4$ R*. Numbers near the traces in D were obtained from the IS + OS with the same number in A and C. (E) Fraction of IS + OS rods out of the total number of rods in 10 representative preparations (S5 Data). (F) Fraction of IS + OS rods from which photoresponses could be measured out of the total number of ISs + OSs in that preparation (S6 Data). These fractions from each preparation were obtained from 10 random fields of view such as those shown in A–C. CaSiR, Calcium ion detecting probe based on silicon rhodamine; IS, inner segment; OS, outer segment. (TIF)

**S2 Fig. Optical manipulation and imaging setup.** 1, inverted microscope; 2, OOT; 3, force measurement module. Optical components: L1, L2, convergent lenses, f1 = f2 = 100 mm; M1, mirror; FTL, fFTL = 55–90 mm; FL, f = 150 mm; DM1 (900 dcsp; Chroma, Bellows Falls, VT, USA); DM2 (XF22045, Chroma); TL; MO, Olympus 60×, NA 1.4, oil immersion; DO, 10×, NA 0.3; DM3 (900 dcsp, Chroma); L3, convergent lens, f = 40 mm. dcsp, Dichroic ShortPass; DM, Dichroic Mirror; DO, condenser objective; FL, Fixed focal Lens; FTL, Focused Tunable Lens; MO, Microscope Objective; NA, numerical aperture; OOT, oscillatory optical trap;

QPD, quadrant photodetector; TL, Tube Lens
(TIF)

**S3 Fig. The effect of repeated mechanical stimulations.** (A) Trapped bead in contact with the base of the rod OS under bright-field IR imaging. (B) Fluorescence change (DF/F) images, showing the ROI (white box) used to quantify the fluorescence change versus time before and during the first mechanical stimulation. (C) Calcium transients evoked by the repeated mechanical stimulations (indicated by the dark arrow). The amplitude of the first and fast calcium transient (indicated by the horizontal red line) is reproducible, while the second and larger component declines. IR, infrared; OS, outer segment; ROI, region of interest
(TIF)

**S4 Fig. Video imaging of the effect of light on the length of rod OS.** (A) A bright-field view of a piece of retina under IR light at 750 nm. (B) Zoom of the yellow dotted box in A. (C) Zoom of the tips of OS in the yellow dotted squares before illumination, during illumination, and after 20 s. The light-induced shortening of the rod OS corresponds to 2–4 pixels: given that a pixel corresponds to approximately 120 nm, the shortening is on the order of 200–400 nm. The enclosed video provides additional support to the light-induced OS shortening. IR, infrared; OS, outer segment.
(TIF)

**S5 Fig. Comparison of the time course of the electrical response measured with suction pipette and shortening measured with OTs.** (A) Upper panel, 3 photoresponses to flashes of light equivalent to about 0.5, 1, and $2.5 \times 10^4$ R* (red, blue, and black traces, respectively); lower panel, time course of shortening evoked by a flash of light at 491 nm, equivalent to about $10^4$ R*. (B) As in A, but on a more expanded timescale; (C) superposition of all these traces after normalization of the maximum to 1. Traces in the upper panel of A were obtained from the same cell, and the trace in the lower panel of B was the average of 5 different experiments to the same light flash in different cells. OTs, optical tweezers.
(TIF)

**S6 Fig. Comparison of the response to a step of light of 20 s duration and equivalent to about 250 Rh*/s in control conditions (black) and in the presence of GsMTx-4 (red) from the same cell.** Maximal photoresponse to a saturating flash of light was 18 pA. GsMTx-4, M-theraphotoxin-Gr1a.
(TIF)

**S7 Fig. Spontaneous calcium transients in the presence of GsMTx-4.** (A) A trapped bead in contact with the rod OS under bright-field IR imaging. (B) Fluorescence change (DF/F) images, showing the 2 ROIs used to quantify the fluorescence change versus time. (C) Time course of the evoked DF/F change from the 2 ROIs in B. Mechanical stimulation (black arrow) as indicated in B. In the presence of GsMTx-4, mechanical stimulation did not evoke any calcium transients, but spontaneous calcium transients could be observed. GsMTx-4, M-theraphotoxin-Gr1a; IR, infrared; OS, outer segment; ROI, region of interest.
(TIF)

**S8 Fig. Syntenic arrangement for a set of 11 families of genes in the vicinity of *TRPC1*, *TRPC4*, and *TRPC5* across 4 species.** Note that only a subset of these families and species was shown in Fig 5. Each panel depicts the arrangement of genes on chromosomes in the named species, and each column depicts a family of paralogous genes. Each row depicts a section of either 1 chromosome or, in several cases, 2 chromosomes, and in one case, 3 chromosomes. Numbers at the ends of each row denote the chromosome, and the numbers under the gene

names give the position of the gene in Mb. For rows that include more than 1 chromosome, dotted vertical bars show breaks between chromosomes, and the chromosome number and gene position are both listed. Expression in photoreceptor classes is indicated by colored shading as follows: red in cones, blue in rods, and gray in both (or uncertain). Gene names are HGNC where present in human, except for the GCs, for which the IUPAR/BPS names are used: GC-E (encoded by *GUCY2D* in human); GC-F (encoded by *GUCY2F*); GC-D (missing from human; olfactory elsewhere). Question mark indicates gene on an unplaced scaffold, and so the presumed chromosomal location is shown. GC, guanylyl cyclase; IUPAR/BPS, International Union of Basic and Clinical Pharmacology/British Pharmacological Society; Mb, megabase; TRPC, transient receptor potential canonical.
(TIF)

**S1 Movie. Video imaging of the effect of light on the length of rod OS, presented in S4 Fig.** The light-induced shortening of the rod OS corresponds to 2–4 pixels: given that a pixel corresponds to approximately 120 nm, the shortening is on the order of 200–400 nm. The enclosed video provides additional support to the light-induced OS shortening. OS, outer segment.
(AVI)

**S1 Data. Original numerical values for Fig 2I.**
(CSV)

**S2 Data. Original numerical values for Fig 2J.**
(CSV)

**S3 Data. Original numerical values for Fig 4D.**
(XLSX)

**S4 Data. Original numerical values for Fig 5.**
(XLSX)

**S5 Data. Original numerical values for S1E Fig.**
(XLSX)

**S6 Data. Original numerical values for S1F Fig.**
(XLSX)

**S1 Raw Image. Raw image of the WB presented in Fig 7E.** WB, western blot.
(PDF)

## Author Contributions

**Conceptualization:** Ulisse Bocchero, Fabio Falleroni, Simone Mortal, Vincent Torre.

**Data curation:** Ulisse Bocchero, Fabio Falleroni, Simone Mortal, Yunzhen Li.

**Formal analysis:** Simone Mortal.

**Investigation:** Ulisse Bocchero, Fabio Falleroni, Yunzhen Li.

**Resources:** Dan Cojoc.

**Writing – original draft:** Ulisse Bocchero, Fabio Falleroni, Simone Mortal, Trevor Lamb, Vincent Torre.

**Writing – review & editing:** Ulisse Bocchero, Fabio Falleroni, Simone Mortal, Trevor Lamb, Vincent Torre.

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
