## [Editor Report · Decision Letter 0]

3 Feb 2020

Dear Dr Mortal, 

Thank you for submitting your manuscript entitled "Rod photoreceptors require mechanosensitivity for optimal phototransduction" for consideration as a Research Article by PLOS Biology.

Your manuscript has now been evaluated by the PLOS Biology editorial staff as well as by an academic editor with relevant expertise and I am writing to let you know that we would like to send your submission out for external peer review.

Please re-submit your manuscript within two working days, i.e. by Feb 05 2020 11:59PM.

Kind regards,

Di Jiang,

Associate Editor

PLOS Biology

---

## [Decision Letter · Decision Letter 1]

28 Feb 2020

Dear Dr Mortal,

Thank you very much for submitting your manuscript "Rod photoreceptors require mechanosensitivity for optimal phototransduction" for consideration as a Research Article at PLOS Biology. Your manuscript has been evaluated by the PLOS Biology editors, an Academic Editor with relevant expertise, and by four independent reviewers.

In light of the reviews (below), we will welcome re-submission of a much-revised version that takes into account the reviewers' comments. We'd like to ask you to address the concerns raised by reviewers 1, 2, and 4 fully, particularly the effect of the 650 nm light exposure on the rods. You will need to respond to reviewer 3's criticisms conscientiously and, as an essential requirement, please comment comprehensively on the quality of your preparation mentioned by this reviewer who points out the fact that frogs rods preparation is very robust. We cannot make any decision about publication until we have seen the revised manuscript and your response to the reviewers' comments. Your revised manuscript is also likely to be sent for further evaluation by the reviewers.

We expect to receive your revised manuscript within 2 months. 

**IMPORTANT - SUBMITTING YOUR REVISION**

*Re-submission Checklist*

*Published Peer Review*

*PLOS Data Policy*

*Blot and Gel Data Policy*

Sincerely,

Di Jiang

PLOS Biology

REVIEWS:

Reviewer #1: This manuscript by Bocchero and colleagues reports novel and interesting investigations of the mechanosensitivity of rod photoreceptors of the marine frog Xenopus laevis. The study combines electrophysiology, biochemistry and optical tweezers to provide evidence for the following observations: 

 (1) The mechanosensitive channels TRPC1 and Piezo1 are expressed in Xenopus rod photoreceptor inner and outer segments. 

(2) Rod receptors (both outer and inner segments) respond to mechanical stimulation by local transient increases in cytosolic calcium.

(3) Inhibitors of these mechanosensitive channels decrease the duration of bright but not dim electrical flash response.

(4) Rod outer segments shorten in response to bright light flashes.

(5) Mechanosensitive channel proteins have significant homology with other protein families known to be involved in vertebrate phototransduction. 

The authors posit that mechanotransduction in photoreceptors is fundamentally important in regulating bright light response kinetics in rod photoreceptors and may have important implication for regulation of outer segment homeostasis by providing mechanisms important for regulation of outer segment renewal. 

The studies are carefully designed, and by their multidisciplinary nature, effectively address the issues of mechanosensitivty of rod photoreceptors. The experiments and data are clearly described and illustrated, and the results that are reported are compelling. Conclusions reached are justified by the nature and quality of the data presented. Of particular importance is the novelty and importance of the findings for basic understanding of vertebrate phototransduction. Specific comments and questions are detailed below:

Page 2, para 2: The authors provide background information in the introduction that the mechanosensitive channels TRPC and Piezo are found in a range of other tissues where they modulate control of osmotic equilibrium and sensitivity to mechanical stimulation. What is missing here is a more complete discussion of the story of TRP channels and their relationship to vision, particularly in invertebrate photoreceptors. Here, these channels have been shown to be calcium permeant, and activated as a part of the PLC/IP3 signaling cascade. Their calcum permeability is likely important as the authors report here that local mechanical stimulation of rods produces calcium transients. The authors should discuss briefly this history, starting in 1969 when Cozens and Manning first reported mutants (TRP) in which flash responses were abnormally transient. Since that time a wealth of electrophysiological, biochemical, and genetic studies have established up to 28 different mammalian isoforms of TRP that regulate a wide range of cellular process that include mechanical sensitivity. 

Page 4, Figure 1 legend: "scale bar 100�m" cannot be correct. Please check and modify

Page 5 para 2: "….basal end appearing dark." I suggest that at an appropriate location in the manuscript (perhaps Methods) there be included a statement regarding the relationship between color in the images of rods and relative calcium concentration. To state that one end of the cell is "dark" is not sufficiently clear.

Page 6, para 2: "DF/F" should be precisely defined (in Methods).

Page 6, para 2 (bottom of page): In the third line from bottom of page calcium is described as being released from internal stores. A few lines later channels are referred to as the means by which cytosolic calcium increases The implicit assumption seems to be that calcium is released from the discs through membrane channels. However, the authors provide no evidence to argue that this is be the case. Could, for instance, the calcium come into the cell from the outside through channels in the plasma membrane?

Page 10, figure 4: "thanc 1s". (?) 

Page 15, para top: The authors present evidence (Figure 6 and legend) that TRPC1 and Piezo 1 channels are found in the inner segment/ellipsoid and data is presented that mechanical stimulation of the IS results in calcium transients. They show also that calcium transients can also be elicited in outer segments, but there is no convincing demonstration that either of these channel types exist there by immunocytochemistry. The argument that some other MSC channel type, as yet unidentified, is responsible, is weak. I am not sure what should be done here to deal with this logical gap.

Page 20, para 2. "…between and Actin 22 and 24 C". Please clarify and/or correct.

P21, para 2. "DF/F" should be defined more precisely.

Page 26 Fig SI2 legend: This legend and its associated figure illustrates the effect of repeated mechanical stimulations. The legend states: "The amplitude of the first and fast calcium transient is reproducible, while the second and larger component declines." On the face of it, this suggests that any mechanical stimulation may cause a change in the cell's response to subsequent mechanical stimulations. What then is the response of the cell to the attachment of a bead? Does this not perturb the channels so as to confuse the effects of stimulations subsequent to bead attachment. Some discussion of this issue would be helpful.

Page 26, para 3 (video imaging): This legend and associated figure describes changes in outer segment length with light stimulation. Was this experiment performed in the presence of the channel inhibitor? If this experiments is described, I missed it. If not, this result would provide an interesting addition 

Reviewer #2: The manuscript by Bocchero and colleagues describes experiments aimed at studying mechanosensitivity in Xenopus rod photoreceptors and its possible link to phototransduction. The authors begin by demonstrating, with the help of a calcium dye and optical tweezers, that mechanical stimulation of isolated functional rods results in local increase in calcium. They next demonstrate that bright flash stimulation in isolated rods causes slight shortening of their outer segments, and that application of the mechanosensitive channel blocker GsMTx-4 produces a slight reversible acceleration of the saturated photoresponse but does not affect the kinetics of the dim-flash response. Finally, the authors perform gene association analysis and find that TRPC1 and Piezo1 are both closely associated with phototransduction proteins and are expressed in the rod inner segments of Xenopus. Based on these results, the authors suggest that mechanosensitivity is required for optimal phototransduction in rods.

The study is original and interesting, and uses state of the art optical tweezers tools. The findings are intriguing but, ultimately, their relevance for normal (or optimal) phototransduction in rods is not very clear. Specific essential issues that should help address this and strengthen the study and its impact include:

- The authors perform a series of negative control experiments to rule out artifacts in their experiments linking calcium and mechanotransduction in rods. However some key control experiments are missing. Specifically:

o It is not shown whether "unresponsive" rods or truncated rod outer segments change their length or calcium levels in response to photostimulation. If mechanosensation is linked to calcium and phototransduction, presumably they should not. Similarly, an important negative control experiment with light-adapted or bleached functional rods is also missing. 

o It is not shown whether application of the MSC blocker blocks the light-induced shortening of the rod outer segments.

o Finally, it is not clear how the authors rule out effects of GsMTx-4 on the mechanosensitive channels from its possible effects on the phototransduction CNG channels of cascade.

- An obvious omission from the study is an experiment on the possible effect of mechanotransduction-induced calcium increase on rod light adaptation. If the light-induced shortening of the rod outer segment is associated with increase in calcium, it should counteract the light-driven decline in calcium in the course of phototransduction, and should therefore, partially suppress light adaptation. A light-adaptation experiment with vs. without the MSC blocker GsMTx-4 might be able to address this question.

- The range of length changes observed in rods is on the order of 100-200 nm - a miniscule shortening effect considering the 50-60 µm length of the rod outer segments in Xenopus. Furthermore, it is not evident that even this small change in length occurs in the intact retina in vivo, when photoreceptors are mechanically bundled together and meshed with the apical processes of the RPE cells in a tightly-packed structure.

Minor issues:

- The authors mention disc shedding in the Introduction and in the Discussion but it is not clear or discussed how this process might be related to mechanosensation.

- The authors should consider showing the effect of the 650 nm light exposure used for calcium imaging on the level of calcium in functional rods (as in Fig 1B), similarly to the effect shown on the saturating photoresponse in Fig. 1F.

- A number of words throughout the MS are split by a hyphen, e.g. pho-ton, re-duction…

Reviewer #3: This ms reports that small mechanical stimulation of the inner segment and outer segments of isolated Xenopus rod photoreceptors induce changes in intracellular calcium and subtle changes in the cells' responses to bright flashes. Because the spider toxin GSMTX-4 abolishes both the force-driven calcium channels and light-induced changes in cell volume, the authors infer that these channels are responsible for both of these phenomena. However, direct evidence that frog rods express any mechanosensitive channels are not convincing. The proximity of TRPC genes to those of other families of phototransduction genes is perhaps interesting in the larger context of the evolution of vision but not relevant to determination of a functional role of these channels in phototransduction. 

More specifically, several different aspects of the manuscript raise questions:

Most (n>40) of the most-nearly intact rods used for calcium imaging had high levels of fluorescence and were unresponsive; far fewer cells (n=10) were deemed functional (p.5). This seems surprising given the typical robustness of rod photoreceptor preparations in cold blooded vertebrates and does not evoke confidence in the preparation.

Local mechanical stimulation using optical tweezers of both inner and outer segments caused transient increases in calcium that in 3 experiments were blocked by GsMTx-4. Although GsMTx-4 is considered a blocker of mechano-sensitive channels it inserts into lipid bilayers rather than blocks these channels directly. Could the calcium transients themselves have no molecular basis in mechano-sensitive channels per se and the toxin's actions be explained by changes in the fluidity of the membranes? Ways to more specifically assess and perturb channel localization and function are needed.

The small light-driven change in outer segment length measured with optical tweezers (Fig. 3) is quite elegant but it appears that the average movement actually slightly precedes light onset (Fig. 3D). For the validation using video imaging (Fig. 3 SI) the change in length is reported to have corresponded to 1-2 pixels or 100-300 nm. How is that possible to see with standard microscopy?

The effect of the spider toxin on the kinetics of the photoresponses is apparent only for the very brightest flashes, and even then the effect is very small. How does this demonstrate a requirement "for optimal phototransduction" (title)? What is the mechanism?

The text descriptions and quality of the data of Fig. 6 are not standard. The staining for Piezo1 in the inner segments is deemed specific but the greater staining of TRPC1 in the outer segments is deemed likely to be autofluorescence. Why do the authors think that, and if there is such evidence for it being an artifact, why show it? I also don't understand the point of Panels D-E - to show lack of TRPC1c staining? If TRPC1 is only in the inner segment and not outer segment, why did the mechanical pulses of Fig. 2 work for both inner and outer segments and be blocked by the spider toxin in both cases? 

Reviewer #4: This manuscript entitled "Rod photoreceptors require mechanosensitivity for optimal phototransduction" by Bocchero et al. demonstrates mechanosensitivity of amphibian (Xenopus) rod photoreceptors and explores its role in the electrical activity of the rods as well as the genetic correlates between the genes coding mechanosensitive channels and phototransduction proteins. The methodology of the paper is solid and well-chosen including a combination of electrophysiology, optical tweezers (OT) and biochemistry. The utilization of OT together with calcium imaging is a clever choice taken into account the resolution needed in revealing the mechanosensitive movements. These are technically challenging recordings that appear to have been very carefully carried out. The overall findings of the paper are novel, have clear merits and are of broad interest to a wide audience in visual neuroscience. I have comments mainly related to the improvement of the presentation.

Overall: The resolution of figures was borderline even for reviewing purposes. 

Figure 1: What is the estimated photoisomerization rate caused by the excitation light (650nm) in these recordings? It would be important to report and/or estimate this for the reader to know the physiological state of the cells in these recordings.

Fig. 1A: It would be good to either show as inset the excitation and emission spectra and/or move the text "CaSiR…Ex:660" etc to the legend of the Fig.

Fig 1B & C: It would be good to show all rods on the same scale: currently there are scalebars (10 um) in each sub panel but the scaling varies a bit (cf. scale bars). The same scaling and a single joint scale bar would be better. 

Page 5: What is the distribution between "response" and "unresponsive" rods? Currently example cells are shown in Fig. 1 but there is no clear statements of the population data related to number of cells etc. It would be useful to understand this distribution. E.g. could you show the distribution of all cells in terms of the a parameter that defines gradient-type "responsiveness" in Fig 1?

Page 6: "semi-dark-adapted": what is the estimated equivalent background light corresponding to these conditions in isomerization rates? This would be very useful to estimate within a reasonable accuracy. "Semi-dark-adapted" is not now anyhow defined.

Fig 3C & D: It would be good to see standard deviation or SEM with the average trace. Now it seems that each individual trace is an average of two traces but it is unclear how many traces are averaged for the "grand" mean and what is the error bar. Shaded SEMs around the "grand" mean would work too.

Fig 4H: What are the units of the ordinate? pA? Please, indicate in the figure.

Page 11 & 15: It is stated that the impact of GsMTx-4 is larger for saturating intensities. It is however unclear if this is simply because the larger the flash strength is the more insomerizations it causes. If this is the case, dim flashes might simply be at the resolution limit of quantification of the effect. In other words, does the effect scale linearly in proportion to the flash strength from dim to intermediate to strong flashes? Is so, there could be a common mechanistic origin for the phenomenon independent of flash strength. This same question is related to the discussion on page 15.

Page 17: It is hard to follow the hypothesis according to which phosphodiesterase could be responsible for the transient shortening of rods. This argument would need to be expanded and/or dropped for clarity of the presentation.

---

## [Editor Report · Decision Letter 2]

28 Apr 2020

Dear Dr Mortal,

Thank you for submitting your revised Research Article entitled "Mechanosensitiviy is an essential component of phototransduction in vertebrate rods" for publication in PLOS Biology. I have now obtained advice from the Academic Editor who has assessed your revisions. We're delighted to let you know that we're now editorially satisfied with your manuscript. 

Before we can formally accept your paper and consider it "in press", we also need to ensure that your article conforms to our guidelines. A member of our team will be in touch shortly with a set of requests. As we can't proceed until these requirements are met, your swift response will help prevent delays to publication. Please also make sure to address the data and other policy-related requests noted at the end of this email.

*Copyediting*

*Published Peer Review History*

*Early Version*

*Submitting Your Revision*

Sincerely,

Di Jiang, PhD

Associate Editor

PLOS Biology

Financial Disclosure: 

-- You declared that "The authors received no specific funding for this work." Please confirm this is indeed the case. 

DATA POLICY:

-- Regardless of the method selected, please ensure that you provide the individual numerical values that underlie the summary data displayed in the following figure panels as they are essential for readers to assess your analysis and to reproduce it: Figures 2IJ, 4D, 5H, 8EF. NOTE: the numerical data provided should include all replicates AND the way in which the plotted mean and errors were derived (it should not present only the mean/average values).

---

## [Editor Report · Decision Letter 3]

26 Jun 2020

Dear Dr Mortal,

On behalf of my colleagues and the Academic Editor, Samer Hattar, I am pleased to inform you that we will be delighted to publish your Research Article in PLOS Biology. 

Early Version

PRESS 

Kind regards,

Alice Musson

Publishing Editor, 

PLOS Biology

on behalf of

Di Jiang, PhD,

Senior Editor

PLOS Biology